# Associations between Perceived Teacher Autonomy Support, Self-Determined Motivation, Physical Activity Habits and Non-Participation in Physical Education in a Sample of Lithuanian Adolescents

**DOI:** 10.3390/bs12090314

**Published:** 2022-08-30

**Authors:** Rasa Jankauskiene, Danielius Urmanavicius, Migle Baceviciene

**Affiliations:** 1Institute of Sport Science and Innovations, Lithuanian Sports University, 44221 Kaunas, Lithuania; 2Department of Physical and Social Education, Lithuanian Sports University, 44221 Kaunas, Lithuania

**Keywords:** self-determination, perceived locus of causality, learning climate, physical education, adolescents, physical activity habits

## Abstract

In this cross-sectional study, we tested the associations between teacher autonomy support, self-determined motivation for physical education (PE), physical activity habits and non-participation in physical education in a sample of adolescents. A total of 715 adolescents (of whom 371 (51.89%) were girls) participated. The ages ranged from 14 to 18 years, with mean ages of 16.00 (SD = 0.79) for girls and 15.99 (SD = 0.75) for boys. The study questionnaire consisted of demographic questions and the Learning Climate Questionnaire, Revised Perceived Locus of Causality in Physical Education Questionnaire, Behavioural Regulation in Exercise Questionnaire 2, Self-Report Habit Index for Physical Activity, Godin Leisure-Time Exercise Questionnaire, Rosenberg Self-Esteem Scale, perceived physical fitness and frequency of non-participation in PE classes. The results showed that perceived teacher autonomy support was directly positively associated with physical activity habits and negatively with non-participation in physical education classes. Autonomous motivation for PE was a mediator between perceived teacher autonomy support and physical activity habits, meaning that higher autonomous motivation was related to higher physical activity habits. Motivation for PE was also a mediator between teacher autonomy support and non-participation in PE. Higher autonomous motivation for PE was associated with less frequent non-participation in PE classes. The findings can inform PE teachers’ practice by showing that supporting students’ autonomy and strengthening their self-determined motivation can facilitate increased participation in PE classes and the formation of students’ physical activity habits.

## 1. Introduction

### 1.1. The Importance of Physical Education for the Lifetime Physical Activity of Adolescents

Modern physical activity (PA) recommendations suggest that school-aged children and adolescents should reduce their amount of sedentary time and engage in an average of 60 min (min)/day of moderate- to vigorous-intensity aerobic PA throughout the week, as well as being regularly involved in strength training exercises [1]. However, substantial numbers of adolescents fail to meet these recommendations [2]. For example, in Lithuania, only 33% of primary school children as well as 30% of adolescents boys and 20% of adolescent girls reach the recommended levels of PA [3]. A major decline in general PA has been observed for the period of adolescence [4]. Declining interest and participation in physical education (PE) classes is a major problem in adolescence, especially in girls [4]. PE is an important subject that might help to enhance engagement in general PA during the day; to provide the possibility of developing; self-concept as well as eudaimonia-based well-being; and to acquire the knowledge, skills, motivation and habits to be active outside school hours and in later life [5]. According to self-determination theory (SDT) [6,7], the quality of motivation is an important agent related to the behavioural, emotional and cognitive outcomes of PE [8,9].

### 1.2. Self-Determination Theory Based Model of Motivational Sequence in the Context of Physical Education

This study is based on SDT as a broad framework explaining human motivation and its related behaviour [7,10,11]. SDT is an organismic theory that assumes that people are inherently prone to psychological growth, integration, learning, mastery and connection with others [11]. SDT postulates the existence of three basic human needs: autonomy, competence and relatedness [12]. Autonomy refers to the need to feel that one’s behaviour is self-determined and that reasons for action are self-endorsed [11]. Competence refers to the need to feel effective and capable of performing various tasks [11,12]. Relatedness denotes the experience of warmth, bonding and care and is satisfied by connecting to and feeling significant to others [11,12]. SDT argues that basic psychological need satisfaction results in greater well-being (meaning in life, life satisfaction, positive affect, self-esteem) and is negatively associated with depression, apathy and anxiety [13]. Satisfaction of basic psychological needs is associated with more positive behavioural, cognitive and achievement outcomes [11,12,13]. According to research in various domains, fulfilling basic psychological needs helps to support self-determined motivation [11,13]. Previous findings in PA, PE and other domains suggested that support for autonomy, competence, and relatedness is associated with greater self-determined motivation [8,13,14].

SDT place types of motivation along a continuum from amotivation to self-determined or intrinsic motivation. The state of amotivation reflects a lack of intentionality and is associated with strong negative behavioural, cognitive, and wellness-related outcomes [11]. External regulation means that person performs activities under the pressure of external factors not related to the activity, such as gaining rewards or good grades, or avoiding punishment, etc. Introjected regulation happens when motivation is partially internalized and activities are performed for internal pressure, because of feelings of anxiety and guilt, or when the individual is seeking to preserve self-esteem (SE). Introjected regulation and external regulation are considered controlled forms of extrinsic motivation. These forms are associated with lower well-being and poorer behavioural, cognitive and achievement outcomes [11]. More autonomous forms of extrinsic regulation are identified and integrated regulations. Identified regulation reflects behaviour that is personally important and valued and the individual experiences a high degree of volition and willingness to act. Integrated regulation occurs when motivation is internalized, and activities assimilate with personal goals, attitudes and values. Finally, intrinsic motivation means that a person’s actions are consistent with self-endorsed reasons for action (e.g., for pleasure, fun, or personal interest). Identified, integrated and intrinsic forms of behavioural regulation are considered as autonomous motivation [8,11].

According to the four-step model of motivational sequence in the context of school PE [8], support for autonomy, competence and relatedness in social environments such as PE leads to an increase in the fulfilment of basic psychological needs at the intrapersonal level [8,15,16,17]. A recent systematic review and meta-analysis demonstrated that PE teachers greatly impact classroom experiences of autonomy and competence [8]. Autonomy-supportive teachers use noncontrolling language, try to understand, acknowledge and be responsive to students’ perspectives and also provide them with rationales and meaningful choices as well as novel tasks [9]. In contrast, controlling teachers are oriented to pressure students to think, feel or behave in particular ways without trying to understand students’ perspectives [8]. Next, research showed that increased students’ satisfaction of basic psychological needs leads to more self-determined students’ motivation in PE [8,16,18,19,20] and leisure time PA [19,20,21]. Competence satisfaction is the most strongly associated with students’ autonomous motivation for PE [17], suggesting that a sense of efficacy and physical competence in physical education is associated with more willing participation in PE [8]. Finally, autonomous students’ motivation for PE further leads to positive affective, cognitive, and behavioural consequences such as higher concentration, positive affect, task challenge, lower unhappiness, greater enjoyment in PE and higher self-esteem [8,16,22,23,24,25].

### 1.3. The Relationships between Self-Determination and Physical Activity Habits

One of the most important tasks of PE is to promote students’ lifetime physical activity [26]. Therefore, researchers paid attention to the processes in which motivation in PA in the PE context is transferred into a leisure time PA context. The trans-contextual model of motivation (TCM) was developed for this purpose. TCM is a multifaceted theory that integrates SDT, the hierarchical model of intrinsic and extrinsic motivation (HMIEM) [27] and the theory of planned behaviour, TPB [28]. The main hypothesis of the model is that students’ perceived autonomy support from their teachers might influence their self-determined motivation in PE, but students’ motivation might be also enhanced in other contexts such as leisure-time PA. Supporting the main tenets of TCM, there is growing evidence that PE environments that promote students’ autonomous motivation for physical education enhance students’ motivation in another context such as leisure time physical activity [18,19,21,22,29,30] and vice versa [31].

However, the strategic aim of PE is not only to promote PA adoption but also to achieve lifetime maintenance of PA. PA maintenance is associated with mechanisms in which PA habits have an important place [32]. The frequency of PA is an important factor of a healthy lifestyle, however, not the frequency of PA per se that is important, but rather, the degree to which the decision to exercise has become a habit [33]. The adoption and maintenance of PA might be explained by modern dual-process theories of self-regulation that postulate explicit (i.e., reflective, deliberate) and implicit (i.e., affective, automatic) processes that responsible for PA adoption and maintenance [34,35]. Automaticity distinguishes habitual behaviour from reasoned actions [36]. It was suggested that habitual behaviours proceed without high cognitive efforts [37] and are performed even in conditions when self-control and motivation are low [37,38].

According to theory and research, a habit might be classified as a behaviour and a psychological construct [32]. Contemporary theory defines a habit as a specific action or behavioural tendency that is enacted with little conscious awareness or reflection in response to a specific set of associated conditions or contextual cues [39]. PA habits comprise multiple sub-actions, some of which may be under deliberate control, and others under automatic control. These behaviours can be initiated and executed habitually [40]. Three elements are important for formation of habits: behavioural repetition, high degree of automaticity and dependence on cues in stable contexts [41]. Habits originate in goal pursuit because people tend to repeat actions that are rewarding or yield desired outcomes [42]. Nevertheless, it has been proposed that habits are distinct from the automatic activation of goals or motivational cues [40]. Prior analyses demonstrated that individuals act in accordance with their habits but not with their primed goals [37].

Habit strength is a continuum, and individuals differ in the extent to which they experience their behaviour as habitual [41,43]. As habits develop, individuals become less sensitive to goals and rewards that previously led to the development of a habit [44]. An important assumption exists that habits might be conceived of as a shift from external goal-dependence to internal goal dependence and internal rewards are necessary for habit formation and maintenance in domains such as PA [45]. Findings of the previous studies suggested that self-determined motivation is associated with PA habits [46,47]. Autonomous motivation might foster the formation of habits directly [46] and indirectly, since autonomous motivation is associated with increased engagement in PA and the latter might promote the development of habits [48]. Finally, autonomous motivation means that individuals engage in PA for enjoyment, fun and inherent interest or that PA is part of an individual’s identity. Previous works revealed that habits develop more quickly if PA and other behaviours are performed for self-determined reasons [42,46]. However, the associations between SDT-based motivational sequence in PE and the strength of PA habits are largely unknown. Since habit formation is thought to aid the maintenance of PA [38], it is important to have more knowledge about the associations between perceived autonomy support, self-determined motivation in PE and strength of PA habits in adolescents. The role of the internal rewards for the formation of stable and persistent behaviour such as PA habits is less explored in PE and more knowledge is needed [44,45]. This knowledge also might help PE teachers to implement strategies that effectively promote the development of PA habits that are necessary for students’ lifetime PA.

### 1.4. Self-Determination in Physical Education and Non-Participation in Physical Education Classes

PA is essential for the health and well-being of adolescents. One recent investigation reported that adolescents across 65 countries who took at least 3 PE classes per week had double the odds of being sufficiently active, with no gender or age differences [49]. Scholars believe that adolescent lifetime PA habits rely on the successful development of physical literacy in PE classes. However, not all students perceive PE as a meaningful practice [50]. One qualitative study concluded that parents might be more supportive of non-participation in PE if they do not believe PE is meaningful and holds value for their children [51]. Analyses of various samples concluded that reasons for non-participation in PE include the use of screen-based activities more than two hours per day, in addition to being female or older than 12 years and overweight, as well as previous negative experiences in PE including teacher support for children who are gifted in sport and increased feelings of incompetence [52,53]. SDT-based works in PE demonstrated that self-determination-based motivation in PE was associated with higher collective engagement in PE [54]. In contrast, controlled motivation is associated with lower-rated collective engagement in PE, higher boredom, and lower achievements in PE [54,55,56]. Thus, in the present study, we expected to confirm the main tendencies of SDT in this sample.

### 1.5. The Present Study

The aim of this study was to explore the associations between teacher autonomy support, self-determined motivation for PE and non-participation in PE, as well as the strength of PA habits, in a Lithuanian sample of adolescents. Based on the main tenets of SDT and previous research in PE, in this investigation, we expected that strength of PA habits would be directly associated with teacher autonomy support and that students’ motivation for PE would mediate it. We also expected that more self-determined motivation would be associated with higher strength of PA habits. Furthermore, we assumed that non-participation in PE classes would be directly and negatively related to teacher autonomy support and that motivation for PE would mediate these associations so that more self-determined motivation would be associated with lower non-participation in PE. The hypothetical model is presented in Figure 1. Additionally, for the purpose of testing concurrent validity of the national language translated instruments, in the present study, we tested associations between perceived autonomy support, self-determined motivation for PE and perceived physical fitness and self-esteem. We expected that these associations would be positive.

## 2. Materials and Methods

### 2.1. Study Participants and Procedure

A total of 715 adolescents (of whom 371 (51.89%) were girls) participated in this study. The children assessed attended 13 different schools in different geographical regions in Lithuania. The participants were in the 9th and 10th grades of conveniently selected public schools. The ages ranged from 14 to 18 years, with a mean age of 16.00 (SD = 0.79) for girls and 15.99 (SD = 0.75) for boys.

The data were obtained during June 2022. The respondents provided their answers by completing a battery of self-report questionnaires designed to measure the study variables. The questionnaires consisted of a set of demographic questions followed by the Learning Climate Questionnaire, Revised Perceived Locus of Causality in Physical Education Questionnaire, Behavioural Regulation in Exercise Questionnaire 2, Self-Report Habit Index for physical activity, Godin Leisure-Time Exercise Questionnaire, Rosenberg Self-esteem Scale, perceived physical fitness and frequency of non-participation in PE classes. This study was approved by the Social Research Ethics Committee of Lithuanian Sports University (Protocol No. SMTEK-113, 10 June 2022). After obtaining permission from the school principals or administrations, the online survey link was circulated with the help of the schools’ PE teachers in the classes of potential research participants. The PE teachers were introduced to the purpose of this analysis and the questionnaire administration protocol. The survey was administered on the SurveyMonkey platform (https://www.surveymonkey.com/) (accessed on 20 June 2022) with an average duration of 25–35 min. Questionnaires were filled out during theoretical PE classes (with no time limit). No information allowing the identification of study participants was collected, and thus, anonymity was ensured. In line with the Declaration of Helsinki ethical and legal principles, the participants were introduced to the aim of this investigation. The participants had the option to agree or refuse to participate in the survey by themselves, with the online form asking “Do you agree to participate in this study?” Those who agreed were provided with the study measures. In cases where a disagreement was provided, the respondents were acknowledged, and the survey was terminated. In addition, there was the ability to stop the survey at any point by closing a browser without recording the answers.

### 2.2. Translation of the Learning Climate Questionnaire (LCQ) and Revised Perceived Locus of Causality in Physical Education Scale (PLOC-R)

The Learning Climate Questionnaire (LCQ) [57] was obtained from the official SDT site (https://selfdeterminationtheory.org/learning-climate-questionnaire/) (accessed on 4 May 2022). The translation of the LCQ into Lithuanian was carefully performed by a professional translator and then translated back into English by another. The final translation was reviewed by an expert in the field of LCQs with professional translators to determine whether the questionnaire covered the concepts it aimed to measure. The original and translated versions are presented in Appendix A, Table A1.

After obtaining the authors’ permission, the instrument Revised Perceived Locus of Causality in Physical Education Scale (PLOC-R) [58] was translated into the Lithuanian language. The translation of this questionnaire was also implemented using the back translation technique. Overall, four experts were involved in the translation process. One English–Lithuanian professional translator translated the scale from English to Lithuanian. These versions were combined and revised. Subsequently, another English–Lithuanian professional translator translated the Lithuanian version back into English. Based on the back-translated English versions, the Lithuanian versions were revised to ensure a comparable meaning of content. Finally, two researchers revised the respective questionnaires in wording and syntax with professional translators, to ensure item clarity and comprehension. A pilot study with 20 boys and 20 girls was conducted and some minimal language—Related corrections were made based on students’ feedback. The original and translated versions are presented in Appendix A, Table A2.

### 2.3. Study Measures

Perceived teacher autonomy support was measured by the LCQ, which contains 15 items with a seven-point Likert scale of response options ranging from “strongly disagree” to “strongly agree” [57]. The questionnaire is typically used with respect to specific learning settings, such as a particular class, at the college or graduate school level. A sample item is “I feel that my teacher of physical education provides me with choices and options”. The negatively formulated statement no. 13 was recoded (1 = 7, 2 = 6, 3 = 5, 4 = 4, 5 = 3, 6 = 2, 7 = 1), and all the response options were averaged, with the final score reflecting a more positive learning climate during PE classes. For this study, the Cronbach’s α was 0.96. As in this investigation the Lithuanian translation of the LCQ was used first, we conducted exploratory (EFA) and confirmatory (CFA) factor analyses. In Appendix A Table A1, the factor loadings and original/*translated* items are presented. As the negatively scored item no. 13 was attributed to a separate factor, we removed it from further analyses. Because a single orthogonal factor was hypothesized, we used the principal axis factoring method with varimax rotation. The Kaiser–Meyer–Olkin (KMO) measure of sampling adequacy was 0.96. A single factor explained approximately 63% of the common variance, and all item-to-factor loadings were satisfactorily high (0.69–0.86). Finally, the fit indices demonstrated satisfactory one-factor fit (CFI = 0.96, root mean square error of approximation (RMSEA) = 0.084 (90% confidence interval (CI) 0.077–0.092), standardized root mean square residual (SRMR) = 0.032).

Motivation for PE was measured using the PLOC-R, which was created on the basis of the SDT and consists of 19 items rated on a seven-point Likert scale ranging from “not true for me” to “very true for me” [58]. The scale comprises five subscales assessing five types of exercise regulation: amotivation, external regulation, introjected regulation, identified regulation, and intrinsic motivation. The participants were asked to assess their motives for participation in PE classes, for example: “I take part in physical education…” “But I really feel I’m wasting my time in PE” (amotivation); “Because in this way I will not get a low grade” (external regulation); “Because I would feel bad about myself if I didn’t” (introjected regulation); “Because it is important to me to be good at sports we practice in PE” (identified regulation); “Because PE is fun” (intrinsic regulation). For this analysis, the Cronbach’s α values for the PLOC-R subscales were as follows: for amotivation—0.83, external regulation—0.63, introjected regulation—0.74, identified regulation—0.80 and intrinsic motivation—0.90. In addition, the Relative Autonomy Index (RAI) is calculated by the equation: (2 × intrinsic motivation) + (1 × identified motivation) + (−1 × introjected motivation) + (−2 × external motivation) + (−3 × amotivation), where a higher score indicates more autonomy in participation in PE classes, and a lower score indicates more controlled regulation and/or amotivation for participation in PE classes [59]. Next, the EFA with varimax rotation revealed a three-factor solution with the eigenvalues > 1 and 59.8% of the variance explained. The factor loadings and translated statements are presented in Appendix A, Table A2. The first factor combined seven statements from the amotivation and external motivation PLOC-R domains, the second included four introjected motivation items, and the third contained eight identified and intrinsic motivation statements. Next, the parallel analysis supported a three-factor solution. Finally, the CFA was run, and the three-factor model demonstrated not excellent, but acceptable fit indices (CFI = 0.91; RMSEA = 0.078 (90% CI = 0.072 − 0.083); SRMR = 0.093). Finally, we tested the three-factor model invariance across the gender groups. The invariance statistics for the configural, metric and scalar models are presented in Appendix A Table A3. Metric (*p* = 0.128) but not scalar invariance (*p* < 0.001) across the gender groups was confirmed.

Motivation for PA was assessed using the Behavioral Regulation in Exercise Questionnaire 2 (BREQ-2) [60]. BREQ-2 is also based on the SDT and comprises the same five subscales representing different levels of autonomy in exercise regulation: amotivation, external, introjected, identified and intrinsic motivation [60]. The questionnaire contains 19 items with the response options on a five-point Likert scale from 1 (“not true for me” up to 5 “very true for me”). The Cronbach’s α values for this investigation were as follows in the same order listed previously: 0.82, 0.86, 0.80, 0.72, and 0.83. For this analysis, we only used the RAI calculated by the equation (−3 × amotivation) + (−2 × external motivation) + (−1 × introjected motivation) + (2 × identified motivation) + (3 × intrinsic motivation), where a higher score indicates more autonomy in exercise regulation, and a lower score indicates more controlled regulation and/or amotivation for exercise [61]. The original five-factor structure of the Lithuanian translation of the BREQ-2 was confirmed in our previous studies in general adult populations [62,63].

The strength of PA habits was measured by the Self-Report Habit Index (SRHI). The SRHI is a 12-item scale designed to measure any habitual behaviours with a seven-point Likert scale [39]. In this study, the SRHI was adapted for physical activity. A sample item is “Physical activity is what I start doing before I realize I’m doing it”. The adequate psychometric properties and unidimensional factor structure were previously confirmed in the Lithuanian general population [64]. In this study, for the SRHI Cronbach’s α was 0.90.

PA was evaluated using the Godin Leisure Time Exercise Questionnaire (LTEQ), measuring individuals’ leisure time exercise including the frequency of mild, moderate, and strenuous exercise at 15 min or more per session over a typical week [65]. The final score was obtained by multiplying the frequency of mild, moderate, and strenuous exercise by 3, 5, and 9 and summarizing the results. A higher score represents a higher level of exercise in terms of frequency and intensity.

The frequency of non-participation in PE classes was assessed with a single question developed for this study: “How often do you skip physical education classes?” The response options were 1—“never”, 2—“rarely”, 3—“sometimes”, and 4—“always”. If PE classes were excused due to health issues and/or doctor’s recommendation, these participants were given a separate response option and further set as missing cases (*n* = 35).

Perceived physical fitness (PPF) was assessed with a single question (“How would you evaluate your own physical fitness when comparing with others?”), developed in our previous study [66]. The response options ranged from 1 (“I am very unfit”) to 5 (“I am very fit”).

Self-esteem (SE) and general feelings of self-worth were assessed with the established Rosenberg Self-Esteem Scale [67]. Participants rated the 10 items (e.g., “On the whole, I am satisfied with myself”) on a four-point Likert scale ranging from 1 (strongly disagree) to 4 (strongly agree). The responses to negative items were recoded (1 = 4, 2 = 3, 3 = 2, 4 = 1), so that the sum of the response options reflected higher SE. In this study, the internal consistency (Cronbach’s α) was 0.87.

### 2.4. Statistical Analysis

Preliminary analyses and correlation analyses, as well as testing the variables’ distribution normality and the internal consistency of the scales, were conducted with Statistical Package for the Social Sciences (SPSS) v.27 (IBM Corp., Armonk, NY, USA). A Cronbach’s α over 0.65 was considered adequate [68], while it should generally be noted that Cronbach’s α values are sensitive to the number of items included in the scale [69]. After confirming the distribution normality of all the continuous variables, the independent-samples t-test was employed to compare the means of the study measures between the boys’ and girls’ groups. Cohen’s d was additionally calculated to represent the effect sizes. Effect sizes above 0.2 were considered small, and those equal to or above 0.5 were considered moderate [70]. Next, the Pearson correlation coefficient was used to test the associations between the variables. Correlations between 0.1 and 0.3 were considered small, those above 0.3 and below 0.5 were considered moderate and those equal to or above 0.5 were considered strong with a significance level of <0.05 [71].

Finally, the EFA, CFA with the multigroup analysis for invariance testing and structural equation modelling (SEM) were run using the Mplus v8.7 (Muthén & Muthén, Los Angeles, CA, USA). The cut-off values for each model fit index were used as recommended by Hu and Bentler: RMSEA ≤ 0.06 for good fit and ≤0.08 for acceptable fit; SRMR ≤ 0.08 for good fit and ≤0.12 for acceptable fit; CFI ≥ 0.95 for good fit and ≥0.90 for acceptable fit [72].

## 3. Results

A comparison of the study measures in boys and girls is presented in Table 1. As expected, LTEQ score, SRHI, PPF and SE mean scores as well as motivation to exercise (RAI from the BREQ-2) were higher in boys than girls. No significant differences were found when contrasting amotivation and introjected regulation with the PLOC-R. In addition, external regulation from the PLOC-R was higher in girls, while the identified and intrinsic types as well as the RAI from the PLOC-R were higher in boys. All differences demonstrated small to medium effect sizes.

Furthermore, the correlations between the PLOC-R subscales, LCQ and the RAI from the BREQ-2 are presented in Table 2. The score of the PLOC-R amotivation subscale correlated positively with external and introjected regulation, and negatively with identified and intrinsic regulation. Positive correlations were observed between the PLOC-R subscales representing greater autonomous motivation for PE, with the strongest magnitude between the identified and intrinsic regulation of 0.75 (*p* < 0.001). Similar trends in the associations were found between the PLOC-R subscales, the LCQ, and the RAI from the BREQ-2: the perceived learning climate and autonomous motivation to exercise correlated positively with more autonomous PLOC-R subscales and negatively with controlled motivation. Introjected regulation from the PLOC-R had a weak positive correlation with the perceived learning climate and negative one with the motivation to exercise.

Table 3 shows the bivariate correlations between the PLOC-R subscales, LCQ and PA, SRHI for PA, frequency of skipping PE classes, PPF and SE. The identified and intrinsic regulation subscale scores from the PLOC-R and the LCQ score exhibited weak to medium positive associations with PA, PA habits, perceived physical fitness and self-esteem. The frequency of skipping PE classes had weak to medium negative relationships with the learning climate and greater autonomous motivation for PE classes subscales. The amotivation and external motivation subscales’ scores from the PLOC-R also demonstrated weak to medium negative relationships with PPF and SE. On the contrary, positive correlations between the greater controlled motivation for PE subscales from the PLOC-R and the frequency of skipping PE classes were found.

Finally, we created a path model based on the SDT. We tested the associations between perceived learning climate, PA habits and frequency of skipping PE classes mediated by motivation for PE. The final path model with the standardized regression weights is presented in Figure 2. It was revealed that perceived teacher autonomy support during PE classes had direct positive effects on PA habits (estimate = 0.15; 95% CI = 0.08–0.22; *p* < 0.001) and motivation for PE (estimate = 0.40; 96% CI = 0.39–0.45; *p* < 0.001), a negative one on the frequency of skipping PE classes (estimate = −0.14; 95% CI = −0.20–(−0.07); *p* = 0.001). Furthermore, there were direct effects from PE motivation to PA habits (estimate = 0.11; 95% CI = 0.04–0.20, *p* = 0.026) and the frequency of skipping PE classes (estimate = −0.24; 95% CI = −0.30–(−0.18); *p* < 0.001). The model demonstrated good fit indices (CFI = 0.998; RMSEA = 0.023 (90% CI = 0.00–0.11); SRMR = 0.012). In addition, configural invariance of the final model across gender groups was tested. Results showed that the model fitted the data across both boys and girls, χ^2^ = 7.526, *p* = 0.376; df = 7; CFI = 0.998; SRMR = 0.030; RMSEA = 0.015 (90% CI = 0.000, 0.068), and the chi-square test difference between unconstrained and constrained models was not significant (*p* = 0.21).

## 4. Discussion

In this study, we investigated the associations between teacher autonomy support, self-determined motivation for PE, the strength of PA habits and non-participation in PE classes in a sample of adolescents. Based on the SDT, we tested a hypothetical model and expected that teacher autonomy support would be directly and positively associated with the strength of PA habits and negatively with non-participation in PE classes. Furthermore, we hypothesized that greater autonomous motivation for PE would be a mediator between teacher autonomy support and PA habits and non-participation in PE. The results confirmed our assumptions and are in line with the main assumptions of SDT [11]. The findings of the present study extend the literature on this topic and provide important new data suggesting that enhanced autonomous motivation in PE context is associated with increased stable and persistent PA habits outside the school. Previous works that applied SDT to PE confirmed that social context and autonomous motivation are associated with positive PE outcomes outside of school. Specifically, autonomous motivation for PE is associated with a higher intention to be physically active and actual reported and objectively measured PA during recess and outside of school [15,18,19,22,29,30,73,74,75]. This research adds important knowledge and goes in line with the knowledge that higher internal rewards (or intrinsic motivation) are associated with the greater habitual behaviour such as PA [44,45]. This is also consistent with previous findings that more autonomous motivation is associated with higher PA habits in adults [46,47]. However, this analysis reports these associations in adolescents and the PE context.

PE teachers greatly impact classroom experiences by fulfilling students’ need for autonomy [8]. Previous studies demonstrated that teacher autonomy support is associated with students’ higher autonomous motivation for PA in leisure time outside of school [21,76] and the perceived controlling behaviour of PE teachers was associated with lower levels of intention to be physically active and lower reported PA outside the school [20,31]. The present study adds new knowledge that perceived autonomy support in social context (teacher autonomy support for PE) is directly associated with behaviour in different context such as habitual PA behaviour outside the school (PA habits). However, the present study is cross-sectional, and the directions of associations are bidirectional, thus we were not able to conclude whether teacher autonomy support and greater autonomous motivation is the cause of higher PA habits and attendance in PE classes. It might be that students with more athletic identities or those already participating in formal or non-formal sports find more enjoyment and fun in PE and perceive that they receive more support from their PE teachers. In contrast, it might be that students who are more sedentary and less gifted in sport feel more controlled by their teachers and find less pleasure, enjoyment and value in PE, and therefore, they do not develop PA habits and report higher levels of non-participation in PE. The nature of this study does not allow us to answer these questions and future ones in other designs should be implemented to further test these assumptions.

The findings of this study are consistent with previous analyses reporting that self-determined motivation for PE is associated with higher engagement in PE and vice versa [54]. This investigation provides a new understanding that lower teacher autonomy support is directly associated with non-participation in PE classes and motivation for PE mediates this association. Thus, it is important to promote teacher autonomy support and self-determined motivation for PE when trying to achieve increased class attendance in PE.

Additionally, in the present study, we hypothesized that autonomy support and self-determined motivation for PE would be associated with greater perceived physical fitness and self-esteem in adolescents. The results of the present study partially supported the hypothesis. Teacher autonomy support and self-determined motivation was associated with higher self-esteem and this finding and go in line with findings of the previous studies [25]. However, perceived physical fitness was not related with teacher autonomy support, but it was associated with more autonomous forms of motivation for PE. This finding overlaps results of previous study in children that showed perceived physical fitness levels were associated with the greater autonomous motivation [77].

The gender analysis showed that boys’ PA, perceived physical fitness, PA habits and self-esteem were higher than girls’. Previous studies reported the same gender tendencies [8,66,78,79]. Boys reported significantly lower non-participation in PE compared to girls, and internal motivation for PA and PE classes was higher than girls. However, we observed no significant gender differences in perceived teacher autonomy support in this sample. This contradicts findings from a prior analysis that reported higher perceived teacher autonomy support for adolescent boys compared with girls [80]. Overall, the results of this investigation suggest that significant differences in PA and internal motivation exist between the genders and boys are more involved and motivated compared to adolescent girls. The reasons why adolescent girls are less motivated for PE and PA might be associated with multiple factors. However, one of the strongest is body image concerns [81]. Body image concerns, body dissatisfaction, low self-esteem are related to higher external exercise motivation [82] and this might be one of the reasons why girls express less autonomous PE and PA motivation. Thus, it is important to pay special attention to adolescent girls and implement specific girls-oriented techniques aiming to foster their enjoyment and inherent interest, as well as positive and functional body image and body satisfaction, with the goal of increasing their internal motivation for PE and strengthening their PA habits [83,84].

One of the goals of this investigation was the translation and testing of the main psychometric properties of Lithuanian versions of the LCQ [57] and PLOC-R [58]. The exploratory factor structure of the LCQ confirmed the one-factor structure and CFA demonstrated satisfactory one-factor fit indices. Furthermore, based on SDT, all the associations between LCQ and other study variables followed the expected directions, confirming the concurrent, discriminant and nomological validity of the instrument. Finally, the internal consistency of the questionnaire was high (Cronbach’s α 0.96). Thus, we can conclude that the Lithuanian version of LCQ is valid and suitable for future studies in adolescent samples. Previous works confirmed the acceptable psychometric properties of the short version of the LCQ [85]. However, to the best of our knowledge, no in-depth psychometric evidence for the full version of the LCQ is available for testing the LCQ after it has been translated to other languages, meaning that comparison is limited.

However, in this sample the original factor structure of the Lithuanian PLOC-R version was not confirmed. EFA revealed the three-factor solution instead of the five-factor version, as in the original structure of the PLOC-R. Identified and intrinsic regulations formed the first factor, items concerning amotivation and external regulation subscales shaped the second factor and introjected regulation formed a separate third factor. The PLOC-R was previously validated in the German and French languages and the original structure of the questionnaire was replicated [86]. Nevertheless, in this analysis, the results of the confirmatory factor structure showed an acceptable three-factor structure. Finally, the full gender invariance of the three-factor structure was not confirmed, so the use of the Lithuanian version of PLOC-R comparing genders is limited. The analysis of the internal consistency of the subscales of the PLOC-R revealed that the external regulation subscale has a Cronbach’s α (0.63), which is too low, and that the Cronbach’s α of introjected regulation meets the minimum requirements for the confirmation of internal consistency (0.74). Therefore, in this study, we only used the RAI index from the PLOC-R, but avoided deeper analyses using subscales of the PLOC-R in the structural mediation model. In this analysis, we were not able to test the effect of separate motivational regulations on the outcomes. However, the correlation analysis of the RAI index from the PLOC-R revealed that associations between the PLOC-R and other variables follow the theory-driven directions. Thus, this might be considered a limitation of this study and future ones using a larger adolescent sample should further test the Lithuanian version of the PLOC-R.

Discussing other limitations of the present study, it is important to address the cross-sectional nature of the research that was discussed previously. Further, we did not assess the basic psychological needs satisfaction of students, so we were not able to measure the full motivational sequence (social context, need satisfaction, motivation and outcomes) proposed by SDT when testing our assumptions [8]. Future works could address this issue. Finally, the generalization of the results of this investigation should be limited, since the sample in this analysis was from Eastern Europe. Future studies should test our findings in other samples.

Finally, based on SDT, future studies might benefit from testing the impacts of basic psychological needs supported PE environments, intrapersonal satisfaction of basic psychological needs and self-determined motivation in PE on PA habit strength and PA habit development in adolescents of various ages. The role of motivation, internal and external PA goals and positive internal reinforcement for the development of PA habits is still not fully understood [44,45]. The results of the present study support the transferability of motivation between PE and other contexts in PA confirming trans-contextual modality of motivation. However, seeking to deeper understand mechanisms of habit formation considering TCM and HMIEM theories might be also beneficial for PE practice and science. The roles of situational, contextual and global factors and these types of motivation on outcomes such as the development of PA habits also need exploration. Future studies of other than cross-sectional designs are necessary to explore these questions.

The present study has important practical implications. The results of this investigation can inform PE teachers’ practice by showing that supporting students’ autonomy and self-determined motivation in PE can facilitate increased participation in PE classes and the formation of students’ PA habits. Further, the results of the present cross-sectional study suggest that adolescents with low PA habits, especially girls, might benefit from increased teacher support for autonomy in PE and increased intrapersonal PE and PA motivation. The outcomes of this examination might also contribute towards PA promotion interventions for adolescents. Providing more intensive autonomy support for adolescents with low PA habits might be an effective strategy in intervention programs aiming to promote healthy lifestyle and long-lasting PA in adolescents.

## 5. Conclusions

Based on SDT, in this study, we tested the associations between teacher autonomy support, self-determined motivation for PE, PA habits and non-participation in PE in a sample of adolescents. The findings of the present study extend the literature on this topic and provides important new data suggesting that enhanced perceived teacher autonomy support and self-determined motivation in PE context is associated with increased stable and persistent PA behaviour (PA habits) outside the school. Findings of the present study also supported transferability of motivation between PE and other contexts in PA confirming trans-contextual modality of motivation. Specifically, the results showed that teacher autonomy support was directly positively associated with PA habits and directly negatively correlated with non-participation in PE classes. Autonomous motivation for PE was a mediator between teacher autonomy support and PA habits, meaning that greater autonomous motivation was related to increased PA habits. Motivation for PE was also a mediator between teacher autonomy support and non-participation in PE. Higher autonomous motivation for PE was associated with less frequent levels of non-participation in PE classes. The findings of this investigation can inform PE teachers’ practice by showing that supporting students’ autonomy and strengthening self-determined motivation can facilitate increased participation in PE classes and strengthen PA habits. Adolescents with low PA habits, especially girls, might benefit from increased teacher support for autonomy in PE and increased intrapersonal PE motivation. Providing more intensive autonomy support for adolescents with low PA habits might be effective strategy in intervention programs aiming to promote healthy lifestyle and lifetime PA in adolescents.

## Figures and Tables

**Figure 1 behavsci-12-00314-f001:**
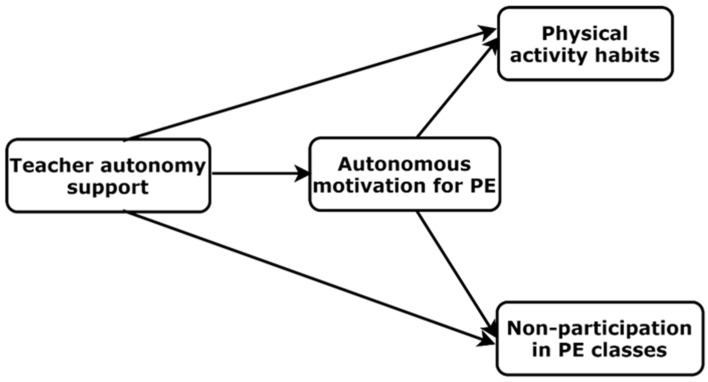
Hypothetical model of the associations between study variables. PE = physical education.

**Figure 2 behavsci-12-00314-f002:**
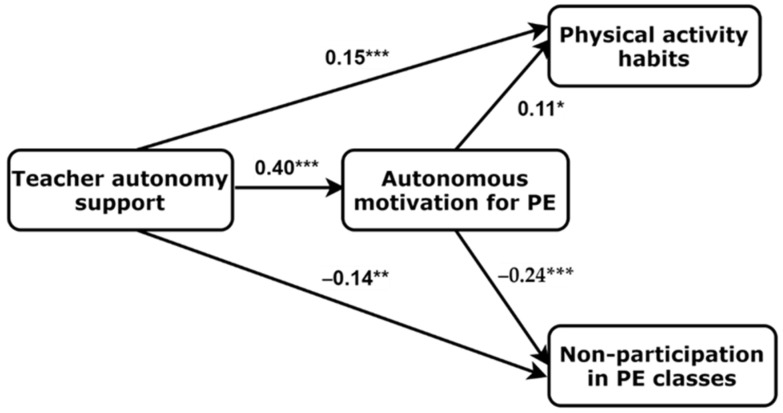
The final model of physical education motivation with the standardized regression weights; * *p* < 0.05, ** *p* < 0.01, *** *p* < 0.001; PE = physical education.

**Table 1 behavsci-12-00314-t001:** Comparison of the study measures in gender groups with calculated effect sizes (*n* = 715).

Study Measures	Range	Boys, *n* = 344		Girls, *n* = 371		Cohen’s d	*p*
		m	SD	m	SD		
Self-Report Habit Index	1–7	4.27	1.27	3.75	1.22	0.42	<0.001
Frequency of NPEC	1–4	1.99	0.88	2.33	0.77	0.41	<0.001
PLOC-R: Amotivation	1–7	2.67	1.55	2.84	1.55	–	0.187
PLOC-R: External regulation	1–7	3.41	1.58	3.74	1.55	0.21	0.004
PLOC-R: Introjected regulation	1–7	3.07	1.52	3.01	1.36	–	0.608
PLOC-R: Identified regulation	1–7	4.26	1.62	3.64	1.50	0.40	<0.001
PLOC-R: Intrinsic regulation	1–7	5.06	1.62	4.39	1.61	0.41	<0.001
RAI from PLOC-R	−32.6–13.7	−3.56	9.54	−6.61	10.33	0.31	<0.001
LCQ	1–7	4.79	1.26	4.77	1.32	–	0.872
Godin LTEQ score	0–395	83.32	54.69	58.35	41.52	0.51	<0.001
RAI from BREQ-2	−14.5–18.7	7.16	6.03	5.73	6.44	0.24	0.002
Perceived physical fitness	1–5	3.52	0.95	3.01	0.86	0.56	<0.001
Self-esteem	10–40	29.56	5.56	27.20	6.62	0.39	<0.001

PE = physical education; LTEQ = Leisure-Time Exercise Questionnaire; NPEC = non-participation in physical education classes; PLOC-R = Revised Perceived Locus of Causality in Physical Education Scale; RAI = Relative Autonomy Index; LCQ = Learning Climate Questionnaire; BREQ-2 = Behavioral Regulation Exercise Questionnaire 2.

**Table 2 behavsci-12-00314-t002:** Correlations between the Revised Perceived Locus of Causality in Physical Education Scale (PLOC-R) subscales, Behavioral Regulation Exercise Questionnaire 2 (BREQ-2) and Learning Climate Questionnaire (LCQ).

Scales and Subscales	m ± SD	Cronbach’s α	AM	EXT	IT	ID	IN	LCQ	BREQ-2
PLOC-R: Amotivation (AM)	2.77 ± 1.55	0.83	1.0						
PLOC-R: External regulation (EX)	3.58 ± 1.58	0.63	0.63 **	1.0					
PLOC-R: Introjected regulation (IT)	3.04 ± 1.44	0.74	0.19 **	0.31 **	1.0				
PLOC-R: Identified regulation (ID)	3.94 ± 1.59	0.80	−0.28 **	−0.16 **	0.49 **	1.0			
PLOC-R: Intrinsic regulation (IN)	4.71 ± 1.65	0.90	−0.47 **	−0.33 **	0.25 **	0.75 **	1.0		
LCQ	4.78 ± 1.29	0.96	−0.31 **	−0.22 **	0.11 *	0.34 **	0.45 **	1.0	
RAI from BREQ-2	6.42 ± 6.29	-	−0.44 **	−0.38 **	−0.20 **	0.30 **	0.42 **	0.27 **	1.0

m = mean; SD = standard deviation; RAI = Relative Autonomy Index; * *p* < 0.05, ** *p* < 0.01.

**Table 3 behavsci-12-00314-t003:** Correlations between the Revised Perceived Locus of Causality in Physical Education Scale (PLOC-R) subscales, Learning Climate Questionnaire (LCQ), physical activity-related behaviours and self-esteem.

Scales and Subscales	PA	SRHI	NPEC	PPF	SE
PLOC-R: Amotivation	0.01	−0.05	0.24 **	−0.06	−0.27 **
PLOC-R: External regulation	0.01	−0.09 *	0.14 **	−0.15 **	−0.28 **
PLOC-R: Introjected regulation	0.10 *	0.08 *	−0.09 *	−0.02	−0.22 **
PLOC-R: Identified regulation	0.20 **	0.34 **	−0.28 **	0.27 **	0.10 **
PLOC-R: Intrinsic regulation	0.17 **	0.33 **	−0.35 **	0.24 **	0.25 **
LCQ	0.02	0.20 **	−0.24 **	0.06	0.25 **

* *p* < 0.05, ** *p* < 0.01; PA = physical activity; SRHI = Self-Report Habit Index for physical activity; NPEC = frequency of non-participation in physical education classes; PPF = perceived physical fitness; SE = self-esteem.

## Data Availability

The dataset generated and analysed during the current study is not publicly available but is available from the corresponding author on reasonable request.

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
