# Peer review of "Associations between Perceived Teacher Autonomy Support, Self-Determined Motivation, Physical Activity Habits and Non-Participation in Physical Education in a Sample of Lithuanian Adolescents"

_behavsci, 2022, doi:10.3390/bs12090314_

Round 1
Reviewer 1 Report
The study is complex, my recommendations are the following:
In the abstract you mention only one questionnaire and in the study there are several, I recommend rewriting the methods section.
Lines 26-28 this conclusion is not clear: The findings can inform, I recommend rewriting.
For keywords, I recommend that you mention in words what PLOC-R represents.
Lines 146-150 I recommend rethinking these sentences, they are unclear, refer to the sample but previously do not mention anything. It is substantiation, not methodology.
Lines 155-159, I recommend rewording.
Line 230, I recommend the correction, Cronbach's α is not for this study, it is for the presented questionnaire.
For the LTEQ questionnaire, do not present Cronbach's α, I recommend the correction.
Author Response
Dear Reviewer,
Thank you for your time reviewing our paper and for your comments. All changes made in the text are highlighted in a blue font.
The study is complex, my recommendations are the following:
In the abstract you mention only one questionnaire and in the study there are several, I recommend rewriting the methods section.
Thank you for this comment. We revised the methods in the abstract, now all questionnaires included in the study are listed.
Lines 26-28 this conclusion is not clear: The findings can inform, I recommend rewriting.
English language was double checked of language experts in the field.
For keywords, I recommend that you mention in words what PLOC-R represents.
Thank you for this comment. We revised key words.
Lines 146-150 I recommend rethinking these sentences, they are unclear, refer to the sample but previously do not mention anything. It is substantiation, not methodology.
We deleted this information from manuscript.
Lines 155-159, I recommend rewording.
This comment was not clear for us. In these lines we present main hypotheses of the research.
Line 230, I recommend the correction, Cronbach's α is not for this study, it is for the presented questionnaire.
Thank you for this comment, corrected.
For the LTEQ questionnaire, do not present Cronbach's α, I recommend the correction.
The Cronbach’s α for the Godin LTEQ cannot be calculated. There are 3 questions in order to assess the number of PA sessions a week of strenuous, moderate and light intensity, so for example study participants can have 0 sessions of strenuous and 20 sessions or light exercise a week.

Reviewer 2 Report
Abstract
Line 17: “queries concerning the learning climate”, if this refers to perceived motivational climate and associated with the autonomy support result, then it should be labelled more specifically.
Line 20: depending on the measure used, “teacher autonomy” should perhaps be termed perceived teacher autonomy, if the adolescents or teachers are being asked about need support provision. If it’s assessed through an observational checklist, then it can stay as it is. Same for line 22.
Introduction
Line 34: Unless journal guidelines state otherwise, physical activity could be abbreviated as it appears frequently through the paper.
Line 49: This statement, as it is written, isn’t particularly accurate. There are numerous studies investigating the association between motivation and PA habits. If you meant it as this blanket statement, then I suggest removing as it isn’t true. However, if the argument made later on is more specific, perhaps it can be moved there and made more detailed after presentation of the literature showcasing how your research will address this gap in the literature base.
Line 58: Generally, when basic needs are presented, they are accompanied by a brief definition so that readers who may not be familiar with SDT will know what they mean.
Line 60: need to explain autonomous motivation, not just what it leads to.
Line 61-63: you do the reverse here and explain what controlled motivation is but do not label it.
Line 64: you introduce intrinsic motivation here but it kind of comes out of nowhere. This whole section on SDT could benefit from a restructured approach. I would suggest introducing the theory generally, then the basic psychological needs and how their satisfaction helps support self-determined motivation (like you have done but with more detail for the BPNs). Then go onto explain that SDT place types of motivation along a continuum from amotivation to intrinsic motivation. Explain each one in order. Then explain that part of this continuum is considered as controlled motivation (external and introjected) and what it leads to, outcome wise, then explain that identified, integrated and intrinsic are considered autonomous motivation and what outcomes that leads to. As you will finish on autonomous motivation, that will lead you smoothly into the next paragraph.
Line 82: This is where you need to be really clear that basic psychological needs can be satisfied (like you’ve outlined above) and supported. You need to delineate more clearly for the reader that BPN satisfaction occurs within the person whereas BPN support occurs in the environment. Providing BPN support is more likely to increase BPN satisfaction and therefore autonomous motivation.
Line 92: “that” should be removed
Line 93: “Finally” is used but then the next sentence starts with “Furthermore”, perhaps these should be swapped around.
Line 96: Ok so here it is clearer about the gap you’re trying to fill so I would suggest removing the sentence picked up in line 49. That being said, there have been studies looking at this, for example:
Barkoukis, V., Chatzisarantis, N., & Hagger, M. S. (2021). Effects of a school-based intervention on motivation for out-of-school physical activity participation. Research Quarterly for Exercise and Sport, 92(3), 477-491.
Jaakkola, T., Washington, T., & Yli-Piipari, S. (2013). The association between motivation in school physical education and self-reported physical activity during Finnish junior high school: A self-determination theory approach. European Physical Education Review, 19(1), 127-141.
Jaakkola, T., Yli-Piipari, S., Barkoukis, V., & Liukkonen, J. (2017). Relationships among perceived motivational climate, motivational regulations, enjoyment, and PA participation among Finnish physical education students. International Journal of Sport and Exercise Psychology, 15(3), 273-290.
Sevil-Serrano, J., Aibar, A., Abós, Á., Generelo, E., & García-González, L. (2022). Improving motivation for physical activity and physical education through a school-based intervention. The Journal of Experimental Education, 90(2), 383-403.
I have the sense that you’re trying to highlight the PA-as-a-habit through the mechanism you describe in this section. However, to be fully clear, it may be better to present the studies like above that have investigated the link between PE motivation and leisure time PA, however, you’re looking from a PA habit point of view. Then clearly explain how this would be more beneficial to the area and from a practical sense for teachers to really round off the argument.
Line 133: the use of “international” doesn’t fully describe that the study investigated 65 countries. You could specify, e.g., “One recent investigation reported that those adolescents, across 65 countries, who took at least…” This is just a suggestion, but I think it’s worth highlighting.
Methods
Line 174: Like before, “learning climate”, should this be perceived motivational climate?
Line 178: links were distributed only to target groups of potential research participants? This needs to be made clearer. The line after makes sense but I’m not sure how this particular line fits in?
Line 197-199: I’m not sure this statement is needed.
Line 214: What do you mean that the items were simplified and adapted according to the youth population based on pre-tests? Does this mean you conducted a pilot trial of the revised instrument? Who determined the simplification? Was this in the language used? Were the number of items reduced? Was the answer format changed? This needs to be detailed more here to ensure a fully trustworthy process.
Line 218: Is the LTEQ reliable in this age group?
Line 224: is this a habit scale for physical activity specifically? Or were 12-items related to PA (but there were more that weren’t included)?
Line231-240: Where do physical fitness and self-esteem fit within the proposed model? I understand why they would be included as variables of interest, but I don’t think these aspects have been clearly set out in the introduction and stand out a little here because of that. You mention self-esteem in relation to introjected regulation in line 77. Fitness is used for the first time in the methods in line 176.
Line 241: What is the numerical value assigned to the answer format?
Line 269: I think it would be of benefit to provide an example item for each type of regulation and label the one example you do have already as intrinsic.
Line 290: what is the answer format for the BREQ-2?
Results
There were two measures for PA (seen in table 1), which one was used in the model? Was it a cumulative score between the two? Or was it PA score in the correlational analysis and habit index in the path analysis? This needs to be clearer.
If gender differences existed on all outcomes in the model except for perceived autonomy support, should there have been two models, one for each gender?
Discussion
Line 400-405: here you may want to refer to the studies I inserted above but differentiate how yours is different.
Line 409-424: Be mindful here with your language use and how far you can extend implications as this is a cross-sectional study. Yes, there is a positive association between perceived autonomy support and PA habits, but we don’t know the direction of this relationship from the study’s findings. This is not to say that what you have said is incorrect, only it may be worth reminding the reader of the cross-sectional nature of the study and what this means. It does set up nicely for future research directions.
Line 432-449: girls were also significantly lower on self-esteem than boys, would that be worth mentioning here and linking to PA, motivation, and literature?
Line 493: insert “satisfaction” after “needs” to be specific.
Line 495: Vallerand’s hierarchical motivational model may be worth a mention here.
The thing missing here is a clear practical implications section for practitioners and researchers: what should teachers do based on these findings and what should researchers do? Some implications are dotted within this section but think the discussion would be strengthened by a definitive section where people can find this information.
Conclusion
Need to highlight the novelty of this work plus clear concrete practical implications for teachers; it reads a little woolly (vague) at the moment.
Author Response
Thank you for your time reviewing our paper and for your valuable comments. All changes made in the text are highlighted in a blue font.
Abstract
Line 17: “queries concerning the learning climate”, if this refers to perceived motivational climate and associated with the autonomy support result, then it should be labelled more specifically.
Thank you for this comment, titles of all questionnaires used are now listed correctly.
Line 20: depending on the measure used, “teacher autonomy” should perhaps be termed perceived teacher autonomy, if the adolescents or teachers are being asked about need support provision. If it’s assessed through an observational checklist, then it can stay as it is. Same for line 22.
Thank you for this comment, corrected.
Introduction
Line 34: Unless journal guidelines state otherwise, physical activity could be abbreviated as it appears frequently through the paper.
We abbreviated term “physical activity” as PA throughout manuscript.
Line 49: This statement, as it is written, isn’t particularly accurate. There are numerous studies investigating the association between motivation and PA habits. If you meant it as this blanket statement, then I suggest removing as it isn’t true. However, if the argument made later on is more specific, perhaps it can be moved there and made more detailed after presentation of the literature showcasing how your research will address this gap in the literature base.
Thank you for this comment. We deleted this statement.
Line 58: Generally, when basic needs are presented, they are accompanied by a brief definition so that readers who may not be familiar with SDT will know what they mean. Line 60: need to explain autonomous motivation, not just what it leads to. Line 61-63: you do the reverse here and explain what controlled motivation is but do not label it. Line 64: you introduce intrinsic motivation here but it kind of comes out of nowhere. This whole section on SDT could benefit from a restructured approach. I would suggest introducing the theory generally, then the basic psychological needs and how their satisfaction helps support self-determined motivation (like you have done but with more detail for the BPNs). Then go onto explain that SDT place types of motivation along a continuum from amotivation to intrinsic motivation. Explain each one in order. Then explain that part of this continuum is considered as controlled motivation (external and introjected) and what it leads to, outcome wise, then explain that identified, integrated and intrinsic are considered autonomous motivation and what outcomes that leads to. As you will finish on autonomous motivation, that will lead you smoothly into the next paragraph.
Thank you for these comments. We addressed all these comments and revised all paragraph following the proposed structure.
Line 82: This is where you need to be really clear that basic psychological needs can be satisfied (like you’ve outlined above) and supported. You need to delineate more clearly for the reader that BPN satisfaction occurs within the person whereas BPN support occurs in the environment. Providing BPN support is more likely to increase BPN satisfaction and therefore autonomous motivation.
We revised this part of text to more clearly present four step sequence of motivation in PE.
Line 92: “that” should be removed
Corrected.
Line 93: “Finally” is used but then the next sentence starts with “Furthermore”, perhaps these should be swapped around.
Corrected.
Line 96: Ok so here it is clearer about the gap you’re trying to fill so I would suggest removing the sentence picked up in line 49. That being said, there have been studies looking at this, for example:
Barkoukis, V., Chatzisarantis, N., & Hagger, M. S. (2021). Effects of a school-based intervention on motivation for out-of-school physical activity participation. Research Quarterly for Exercise and Sport, 92(3), 477-491.
Jaakkola, T., Washington, T., & Yli-Piipari, S. (2013). The association between motivation in school physical education and self-reported physical activity during Finnish junior high school: A self-determination theory approach. European Physical Education Review, 19(1), 127-141.
Jaakkola, T., Yli-Piipari, S., Barkoukis, V., & Liukkonen, J. (2017). Relationships among perceived motivational climate, motivational regulations, enjoyment, and PA participation among Finnish physical education students. International Journal of Sport and Exercise Psychology, 15(3), 273-290.
Sevil-Serrano, J., Aibar, A., Abós, Á., Generelo, E., & García-González, L. (2022). Improving motivation for physical activity and physical education through a school-based intervention. The Journal of Experimental Education, 90(2), 383-403.
Thank you, text is revised, and these important articles are included and cited in manuscript.
I have the sense that you’re trying to highlight the PA-as-a-habit through the mechanism you describe in this section. However, to be fully clear, it may be better to present the studies like above that have investigated the link between PE motivation and leisure time PA, however, you’re looking from a PA habit point of view. Then clearly explain how this would be more beneficial to the area and from a practical sense for teachers to really round off the argument.
More arguments why it is important to study PA habits based on SDT are added to manuscript.
Line 133: the use of “international” doesn’t fully describe that the study investigated 65 countries. You could specify, e.g., “One recent investigation reported that those adolescents, across 65 countries, who took at least…” This is just a suggestion, but I think it’s worth highlighting.
Thank you, corrected.
Methods
Line 174: Like before, “learning climate”, should this be perceived motivational climate?
Thank you for this comment, titles of all questionnaires used are now listed correctly.
Line 178: links were distributed only to target groups of potential research participants? This needs to be made clearer. The line after makes sense but I’m not sure how this particular line fits in?
Thank you for this comment. The sentences were revised removing redundant information.
Line 197-199: I’m not sure this statement is needed.
Thank you for this comment. The statement was removed.
Line 214: What do you mean that the items were simplified and adapted according to the youth population based on pre-tests? Does this mean you conducted a pilot trial of the revised instrument? Who determined the simplification? Was this in the language used? Were the number of items reduced? Was the answer format changed? This needs to be detailed more here to ensure a fully trustworthy process.
Thank you for this valuable comment.
Some minimal language - related changes were made after the pilot study with 20 boys and 20 girls according to study participants’ comments. Neither the number of statements nor the response options or content were changed. We have added the clarification to the text.
Line 218: Is the LTEQ reliable in this age group?
Thank you for this comment. The Godin LTEQ was initially developed for young students aged 7-9 years, later it was started to use for older students and adults.
Godin, G. (1983). Psychosocial factors influencing intentions to exercise of young students. Graduate Department of Community Health, University of Toronto, Toronto.
Godin, G. (2011). The Godin-Shephard leisure-time physical activity questionnaire. Health & Fitness Journal of Canada, 4(1), 18-22.
Line 224: is this a habit scale for physical activity specifically? Or were 12-items related to PA (but there were more that weren’t included)?
Thank you for this comment. The Self-Report Habit Index is designed to measure any habitual behaviours (washing hands, smoking, marijuana use, etc.). In our study all 12 items were used to measure habitual behavior only in physical activity, for example:
Behavior X is something . . . (where X can be any habitual behavior)
- I do frequently.
- I do automatically, etc.
Verplanken, B.; Orbell, S. Reflections on past behavior: A self-report index of habit strength. J Appl Soc Psychol. 2003, 33, 1313–1330.
We added necessary information in Methods section presenting SRHI.
Line231-240: Where do physical fitness and self-esteem fit within the proposed model? I understand why they would be included as variables of interest, but I don’t think these aspects have been clearly set out in the introduction and stand out a little here because of that. You mention self-esteem in relation to introjected regulation in line 77. Fitness is used for the first time in the methods in line 176.
Thank you for this comment. In this study, perceived physical fitness and self-esteem were used to support the validity of the newly translated instruments. However, we included more information in Introduction presenting importance of perceived competence in PA for intrinsic motivation.
Line 241: What is the numerical value assigned to the answer format?
Thank you for this comment. The numerical values are provided.
Line 269: I think it would be of benefit to provide an example item for each type of regulation and label the one example you do have already as intrinsic.
Thank you for this comment. Sample items are provided for each type of regulation.
Line 290: what is the answer format for the BREQ-2?
Thank you for this comment. Answers’ format for the BREQ-2 was provided.
Results
There were two measures for PA (seen in table 1), which one was used in the model? Was it a cumulative score between the two? Or was it PA score in the correlational analysis and habit index in the path analysis? This needs to be clearer.
We corrected Table 1. For the model, Self-Report Habits Index for the strength of PA habits was used.
If gender differences existed on all outcomes in the model except for perceived autonomy support, should there have been two models, one for each gender?
Thank you for this comment. Despite observed differences in gender groups, the strength of the associations between study measures was similar in boys and girls. Separating the path analyses into gender groups only resulted in a power loss. To support the decision to keep one model for the total sample, we have added information about the model configural invariance across gender groups.
Discussion
Line 400-405: here you may want to refer to the studies I inserted above but differentiate how yours is different.
We cited important studies and revised the paragraph providing more information on the novelty of the study.
Line 409-424: Be mindful here with your language use and how far you can extend implications as this is a cross-sectional study. Yes, there is a positive association between perceived autonomy support and PA habits, but we don’t know the direction of this relationship from the study’s findings. This is not to say that what you have said is incorrect, only it may be worth reminding the reader of the cross-sectional nature of the study and what this means. It does set up nicely for future research directions.
Thank you for this comment. We revised this paragraph and included discussion in it concerning limitations related to cross – sectional nature of study.
Line 432-449: girls were also significantly lower on self-esteem than boys, would that be worth mentioning here and linking to PA, motivation, and literature?
We included information on lower self-esteem in girls and linked PA, self-esteem, motivation, and literature.
Line 493: insert “satisfaction” after “needs” to be specific.
Corrected.
Line 495: Vallerand’s hierarchical motivational model may be worth a mention here.
The thing missing here is a clear practical implications section for practitioners and researchers: what should teachers do based on these findings and what should researchers do? Some implications are dotted within this section but think the discussion would be strengthened by a definitive section where people can find this information.
Thank you for this comment. We included information concerning future directions for similar studies in one paragraph and practical implications in other paragraph at the end of the discussion.
Conclusion
Need to highlight the novelty of this work plus clear concrete practical implications for teachers; it reads a little woolly (vague) at the moment.
We included statements on the novelty of our work and inserted more concrete practical implications.

Round 2
Reviewer 2 Report
Thank you for addressing the comments I made in the first round. I have a few minor comments, and one comment remaining from the last review.
Line 78: Missing full stop after “…outcomes [11]”
Line 88: “the” and “in” need to be swapped.
Line 106: Not just recently but over time.
Line 124: “are” may be replaced by “that” to read better.
Line 143: “An important assumption exists”
Physical fitness and self-esteem still remain a bit of a mystery when it comes to later on in the paper. They are briefly mentioned in the introduction but not in their importance, so when data is collected and offered, it comes as a surprise for the reader. They don’t appear in “the present study” section or the hypotheses, nor the model. It needs to be more clearly explained in the introduction, the importance of self-esteem and physical fitness in general, as well as in relation to motivation. They need to be present in the aims and the hypothesis, if they are to be kept in.
Line 302: Is it “not truth for me” or “not true for me”?
Line 433: should it be “hypothesised” rather than “assumed”?
Author Response
Thank you for the valuable comments. All corrections in the manuscript are marked in blue.
Line 78: Missing full stop after “…outcomes [11]”
Corrected
Line 88: “the” and “in” need to be swapped.
Corrected
Line 106: Not just recently but over time.
Corrected
Line 124: “are” may be replaced by “that” to read better.
Corrected
Line 143: “An important assumption exists”
Corrected
Physical fitness and self-esteem still remain a bit of a mystery when it comes to later on in the paper. They are briefly mentioned in the introduction but not in their importance, so when data is collected and offered, it comes as a surprise for the reader. They don’t appear in “the present study” section or the hypotheses, nor the model. It needs to be more clearly explained in the introduction, the importance of self-esteem and physical fitness in general, as well as in relation to motivation. They need to be present in the aims and the hypothesis, if they are to be kept in.
Thank you for this valuable comment. Variables of perceived physical fitness (PPF) and self-esteem (SE) were included in the present study as measures helping to test the concurrent validity of national language-translated instruments. Therefore, we did not include them in the general aim of the study nor in the model. We avoided a deep presentation of these variables in the Introduction since the length of the article is limited. However, we added additional information in the Introduction‘s subsection „The present study“ explaining why these variables are included in the study. We also changed the order of variables in Table 1 presenting main variables (variables in the model) in the first positions and keeping additional variables (PPF and SE) at the end of the table. Also, we added a brief paragraph discussing findings in the Discussion section.
Line 302: Is it “not truth for me” or “not true for me”?
Corrected
Line 433: should it be “hypothesised” rather than “assumed”?
Corrected
